# VideoMolmo: Spatio-Temporal Grounding Meets Pointing

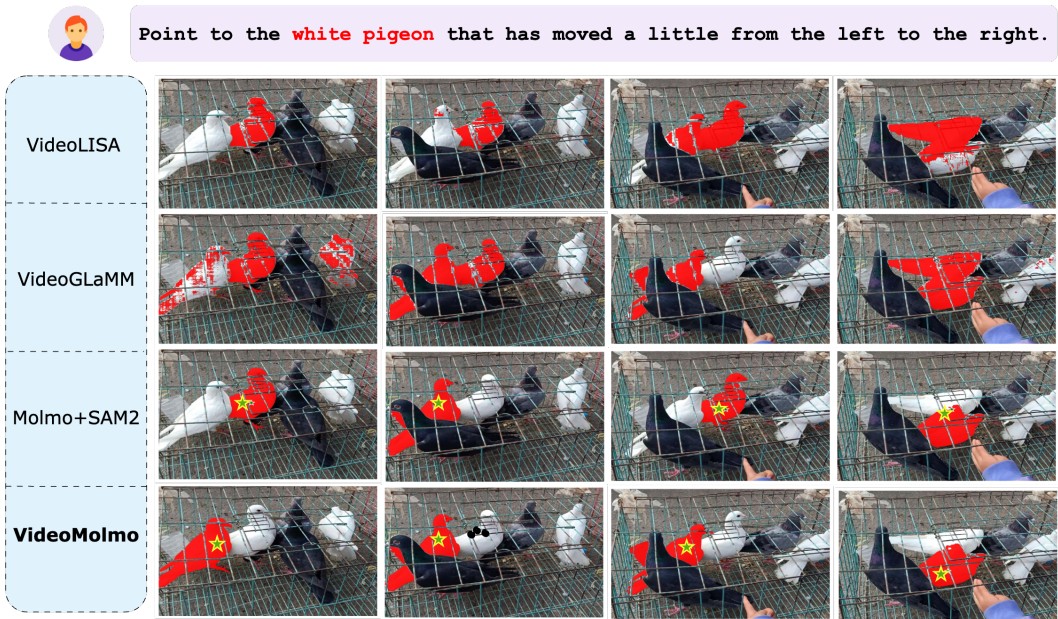

Figure 1: Given complex referring expressions, VIDEOMOLMO improves spatio-temporal reasoning in visual grounding by decomposing the task into sequential steps—pointing (star) followed by mask generation (red)—yielding more accurate and coherent segmentations than prior approaches.

## ABSTRACT

Spatio-temporal localization—the ability to identify both the position and temporal evolution of objects—is essential for applications from cell tracking to autonomous navigation. Recent Video Large Multimodal Models (Video-LMMs) show promise but remain limited by coarse predictions, heavy reliance on dense mask optimization, and limited interpretability. We introduce VIDEOMOLMO, a two-stage framework that grounds objects through point-based localization. Rather than directly predicting dense masks, VIDEOMOLMO first produces precise points as lightweight, interpretable anchors, which are then used for downstream tasks including referring segmentation, video object segmentation, and counting. By decoupling localization from task execution, our approach provides more robust and transparent reasoning. Built on Molmo, our framework incorporates a temporal attention module for cross-frame reasoning and introduces a novel bidirectional temporal mask fusion strategy, enabling coherent point propagation and accurate segmentation. To facilitate training and evaluation, we release a large-scale spatio-temporal pointing dataset of 72k video–caption pairs with 100k annotated points and curate VPoS-Bench, a challenging benchmark spanning five real-world domains. Experiments show that VIDEOMOLMO outperforms existing approaches, with gains of $5.4$ percentage points (pp) on VPoS-Bench and $9.5$ pp on MeViS. This highlights the effectiveness of point-based representations as a foundation for interpretable, fine-grained reasoning in dynamic visual environments.

# 1 INTRODUCTION

Understanding dynamic visual scenes requires not only recognizing what objects are present, but also tracking where they are and how they move over time. This ability, known as spatio-temporal localization, is essential for many applications: biologists follow cell trajectories in microscopy videos, autonomous vehicles monitor pedestrians and traffic, and robots interact with moving objects in cluttered environments.

Recent progress in Video Large Multimodal Models (Video-LMMs) (Maaz et al., 2024; Li et al., 2023; Lin et al., 2023; Munasinghe et al., 2024; Bai et al., 2024) has opened the door to solving such tasks from natural language queries. These models can, e.g., answer questions like "Which player in the red jersey passed the ball?" by linking language with the video content. While impressive, current approaches still struggle in the very settings where precise reasoning matters most.

Consider Fig. 1, where the query is to find "the white pigeon that has moved slightly from left to right" among several pigeons. Solving this requires two things: (i) temporal reasoning to notice subtle movement across frames, and (ii) fine-grained localization to pinpoint the correct individual bird. Current Video-LMMs (Munasinghe et al., 2024; Bai et al., 2024) often fail on such tasks: they either predict multiple objects when only one is correct, or they highlight regions too coarsely to be useful. These errors stem from how existing models are trained i.e., directly predicting dense segmentation masks from video and text.

This design creates fundamental problems. First, dense masks make it difficult to interpret what the model "reasoned" before producing an output, since there is no explicit intermediate signal. Second, the optimization is unnecessarily complex: predicting every pixel's label is heavy-handed when many real-world tasks only need precise object localization. Third, the outputs are often noisy or coarse, since refining boundaries at pixel-level precision is especially hard in dynamic videos.

We argue that points, rather than dense masks, provide a better foundation for video grounding. A single point can precisely indicate an object of interest, even under occlusion, without the overhead of pixel-by-pixel predictions. Points are also interpretable (it is clear what the model meant) and versatile, since they can serve as prompts for downstream tasks like segmentation, tracking, and counting.

Building on this insight, we introduce VIDEOMOLMO, a novel two-stage framework for language-guided video grounding. In the first stage, the model predicts points that represent object identity and location across time. In the second stage, these points guide task-specific modules—such as referring segmentation, video object segmentation, and counting. By separating reasoning (where is the object?) from execution (how to use this location for a downstream task), our approach is both more interpretable and more robust than end-to-end dense prediction.

VIDEOMOLMO builds on Molmo (Deitke et al., 2024) by incorporating a temporal attention module designed to explicitly capture cross-frame dependencies. For segmentation, we propose a temporal mask fusion pipeline, which efficiently propagates point predictions bidirectionally across frames to produce temporally consistent masks. To support training, we release the first large-scale dataset for spatio-temporal pointing, consisting of 72k video–caption pairs and 100k annotated object points. We further establish VPoS-Bench, a challenging benchmark spanning five diverse scenarios: cell tracking, egocentric vision, autonomous driving, video-GUI interaction, and robotics.

Extensive experiments demonstrate that VIDEOMOLMO consistently outperforms existing approaches. On VPoS-Bench, it improves by 5.4 percentage points (pp) over the strongest baseline, and on the MeViS referring segmentation benchmark (Ding et al., 2023), it outperforms prior models by 9.5 pp, despite never being trained on dense masks. These results highlight that point-based reasoning provides a powerful and generalizable foundation for fine-grained video understanding.

# 2 RELATED WORK

**Video-LMMs.** LMMs such as (Liu et al., 2023; Zhu et al., 2023a) have demonstrated notable advancements due to their strong zero-shot abilities, made possible because of their training on millions of image–text pairs. Typically, such models project visual information into the latent space of an LLM via an encoder and a connector, thereby aligning the information from vision and text

modalities. Work on LMMs paved the way for the development of Video-LMMs (Li et al., 2023; Zhang et al., 2023b; Lin et al., 2023; Maaz et al., 2024; Wang et al., 2024; Zhu et al., 2025; Bai et al., 2025; Zhang et al., 2025), which, unlike image-based LMMs, can reason about dynamic video content. While effective for overall video input comprehension, these methods fall short of fine-grained visual grounding in videos.

**Visual grounding.** Recent works in Grounded LMMs (Rasheed et al., 2023) have sparked tremendous interest among the research community. Visual grounding (Liu et al., 2021) seeks to identify the location of nouns or short phrases (such as a man with a blue shirt) within an image. These models are trained on large datasets of image–caption pairs along with dense segmentation masks associated with the objects in the caption. (Bai et al., 2024; Munasinghe et al., 2024) extended grounding to video data by releasing a large dataset of grounded video-QA pairs along with the masks associated with the objects. Training on such large video-grounded datasets allowed for video grounding. In contrast, our VIDEOMOLMO model and dataset focus on predicting precise object-level points, a lightweight representation essential for tasks such as autonomous driving, counting, and robotics.

**Language-assisted object tracking.** Most text-based tracking methods are limited to tracking a single object only (Yang et al., 2020; Zhao et al., 2023a; Wang et al., 2021; Li et al., 2022). However, real-world applications can feature multiple object trajectories, making it challenging for single-object tracking methods. (Nguyen et al., 2023) propose Type-toTrack along with a tracking dataset 'GroOT' for multi-object tracking. However, they track objects via bounding boxes and not precise points, which limits their applicability. Another work, SAM-PT (Rajič et al., 2025), proposes using the SAM (Ravi et al., 2024) model along with a long-term point tracking mechanism for point-centric interactive video segmentation. However, since their method adapts a 2D model to handle video data, it faces challenges in temporal consistency, especially in cases of occlusion and fast-moving objects. In contrast, our proposed VIDEOMOLMO is trained end-to-end on our training dataset and maintains temporal consistency via a dedicated memory module.

## 3 VIDEOMOLMO

**Task Definition.** Given an input video $V \in \mathbb{R}^{|\mathcal{T}| \times H \times W \times C}$, where $H$, $W$, and $C$ denote the height, width, and number of channels of each frame, and $|\mathcal{T}|$ is the number of frames, together with a textual query $\mathcal{X}$, the goal is to predict a set of points $\mathbb{P} = \{\mathbb{P}_t\}_{t=1}^{|\mathcal{T}|}$ for every object referred to by $\mathcal{X}$ in each frame of the video. Each $\mathbb{P}_t = \{(x_i^t, y_i^t)\}_{i=1}^{\mathcal{O}_t}$ represents a set of $\mathcal{O}_t$ two-dimensional coordinates in frame $t$ that localize the objects described in $\mathcal{X}$. These predicted points provide a lightweight and interpretable representation that can be directly leveraged for downstream tasks such as grounding, segmentation, and counting.

## 4 ARCHITECTURE

VIDEOMOLMO extends Molmo (Deitke et al., 2024) from static image understanding to spatio-temporal video grounding. The framework consists of four end-to-end trainable components: (1) a visual encoder, (2) a temporal module, (3) a visual projector, (4) a decoder-only LLM. Additionally, we incorporate a frozen bidirectional temporal mask fusion module to convert point predictions into consistent masks (Fig. 2).

For each video frame $\mathcal{T}_i \in V$, we generate $N$ overlapping spatial crops to capture both global context and fine-grained details. Each crop is independently encoded by a visual backbone $\mathcal{F}$. Following (Deitke et al., 2024), we build patch features by concatenating representations from the third-to-last and tenth-to-last ViT layers. The resulting feature for crop $j$ of frame $\mathcal{T}_i$ is $f_{\mathcal{T}_i}^j \in \mathbb{R}^{P \times D}$, where $P$ is the number of patches and $D$ is the feature dimension. Since the visual backbone processes 2D frames independently, we introduce a dedicated temporal module $\mathcal{M}$ to model cross-frame dynamics. For frame $\mathcal{T}_i$, we aggregate features from the past $l$ frames $\{\mathcal{T}_{i-l}, \ldots, \mathcal{T}_{i-1}\}$ and compute their mean $f_{\mathcal{T}_{i-}^*}$. The temporal module then refines the current frame features by incorporating this historical context:

$$\hat{f}_{\mathcal{T}_i} = f_{\mathcal{T}_i} + \mathcal{M}(f_{\mathcal{T}_i}, f_{\mathcal{T}_{i-}^*}). \tag{1}$$

To enable joint spatio-temporal reasoning, we reshape the frame features $f_{\mathcal{T}_i} \in \mathbb{R}^{N \times P \times D}$ into local patch windows of shape $\mathbb{R}^{N \cdot (P/4) \times 4 \times D}$. This allows $\mathcal{M}$ to exchange information across neighboring

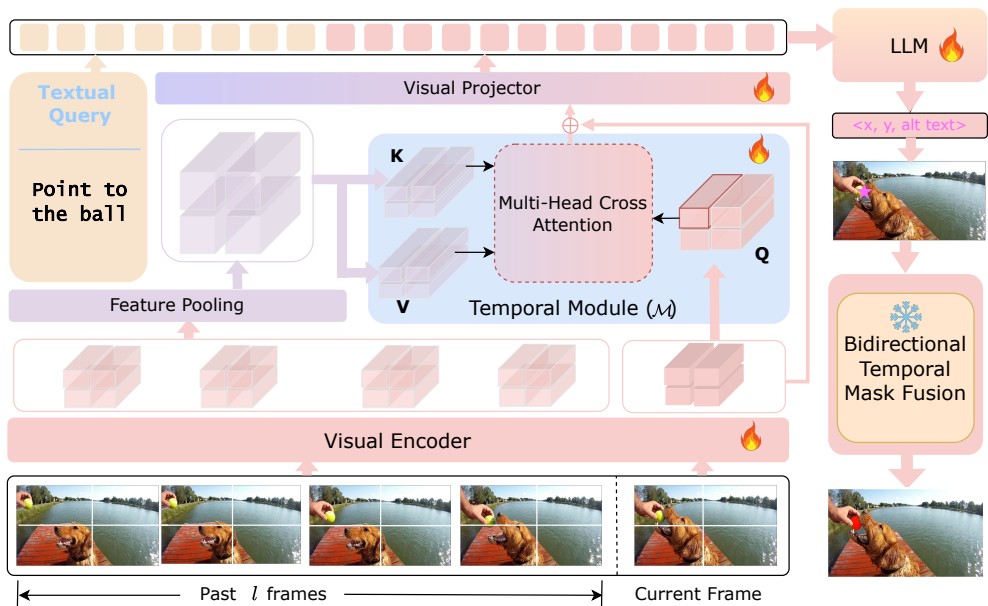

Figure 2: **VIDEOMOLMO Architecture.** The visual encoder extracts multi-crop features from the current frame and the past $l$ frames. These are processed by the *Temporal Module* $\mathcal{M}$ via multi-head cross-attention, where the query comes from the current frame, and key and value from the mean of previous frames. The output is fused with the original features to enrich temporal cues while preserving the spatial details of the current frame. The combined visual-textual representations are then passed to the LLM to predict grounded points. These points are converted into masks using our *Bidirectional Temporal Mask Fusion* module, ensuring temporally consistent pixel-level grounding.

patches while propagating temporal context. The temporally-enriched features $\hat{f}_{\mathcal{T}_i}$ are pooled using attention into a frame-level representation, then projected into the language embedding space via projector $\mathcal{P}$. This visual representation is concatenated with the tokenized query $q$ and fed to the decoder-only LLM, which autoregressively generates grounded point coordinates $(x,y)^{\mathcal{O}}$, where $\mathcal{O}$ is the number of localized objects:

$$p = \text{LLM}\big([\mathcal{P}(\hat{f}_{\mathcal{T}_i}); q]\big) \tag{2}$$

The model is trained end-to-end using a cross-entropy loss between predicted text **p** and ground-truth one-hot labels $\mathbf{p}^*$ in an autoregressive manner:

$$\mathcal{L}_{CE} = -\mathbf{p}^* \cdot \log(\mathbf{p}) \tag{3}$$

## 4.1 TEMPORAL MODULE

The original Molmo architecture (Deitke et al., 2024) was developed for static images and cannot model the temporal dynamics inherent in video data. To address this, we introduce a novel dedicated *temporal module* $\mathcal{M}$ that infuses each frame with temporal context from the preceding $l$ frames, inspired by prior approaches (Nguyen et al., 2023; Lai et al., 2020). For each crop $j$ of frame $\mathcal{T}_i$, patch features $f_{\mathcal{T}_i}^j \in \mathbb{R}^{P \times D}$ are extracted. These features are then flattened across all $N$ crops to obtain vectors $f_{\mathcal{T}_i}$ and $f_{\mathcal{T}_{i-}^*} \in \mathbb{R}^{(N \cdot P) \times D}$, where the latter represents context features aggregated from the preceding frames . To capture fine-grained temporal correspondences, we apply multi-head cross-attention (MHCA) over the patch features, where the query comes from the current frame $f_{\mathcal{T}_i}$, and the key and value come from the context features $f_{\mathcal{T}_{i-}^*}$. Formally, $\texttt{MHCA}\left(f_{\mathcal{T}_i},\ f_{\mathcal{T}_{i-}^*},\ f_{\mathcal{T}_{i-}^*}\right)$ denotes the final temporally attended features. By restricting cross-attention to local neighborhoods rather than global interactions, the module preserves spatial locality while amplifying subtle motion cues that would otherwise be lost in coarse global pooling.

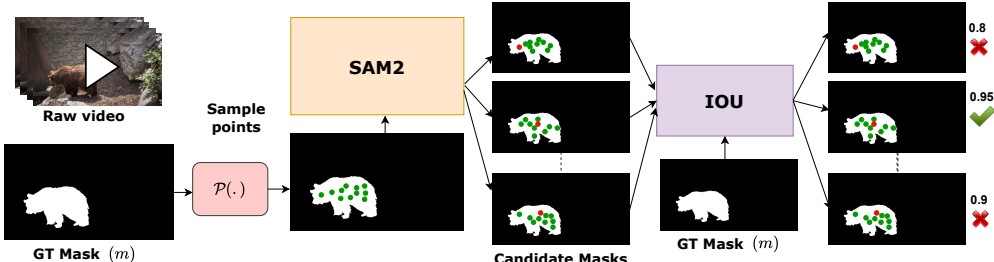

Figure 3: **VIDEOMOLMO annotation pipeline:** We construct point-level supervision from frame-level masks using a semi-automatic process. For each frame, $k$ points are sampled on the mask and passed to SAM2 (Ravi et al., 2024) to generate candidate masks. The point with the highest-IoU candidate mask (with respect to the ground truth) is selected as the optimal annotation.

## 4.2 BIDIRECTIONAL TEMPORAL MASK FUSION

While VIDEOMOLMO predicts grounded point coordinates corresponding to objects mentioned in the textual query, most existing evaluation protocols are designed to operate on segmentation masks. Therefore, for compatibility and consistent evaluation, we introduce *Bidirectional Temporal Mask Fusion*, a novel post-processing technique that leverages SAM2 (Ravi et al., 2024) to convert points to dense masks.

**Sparse Point-to-Mask Conversion.** Instead of running dense inference across all frames, we sparsely sample frames at a rate $k$. For two consecutive sampled frames $\mathcal{T}_i$ and $\mathcal{T}_{i+k}$, the predicted points are converted into masks $m_i$ and $m_{i+k}$ using SAM2 (Ravi et al., 2024).

**Bidirectional Propagation.** For intermediate frames $\mathcal{T}_{i+n}$ ($0 < n < k$), we propagate masks from both directions:

$$
\begin{aligned}
\widehat{m}_{i+n}^{\rightarrow} &= \texttt{Prop}^{\rightarrow}(\{\mathcal{T}_i, m_i\}, \mathcal{T}_{i+n}), \\
\widehat{m}_{i+n}^{\leftarrow} &= \texttt{Prop}^{\leftarrow}(\{\mathcal{T}_{i+k}, m_{i+k}\}, \mathcal{T}_{i+n})
\end{aligned}
\tag{4}
$$

where $\widehat{m}_{i+n}^{\rightarrow}$ and $\widehat{m}_{i+n}^{\leftarrow}$ are masks propagated from past and future, respectively.

**Fusion Strategy.** To reconcile the two estimates, we compute their IoU:

$$
m_{i+n} = \begin{cases} \widehat{m}_{i+n}^{\rightarrow} \cap \widehat{m}_{i+n}^{\leftarrow}, & \text{if } \texttt{IoU} \geq \tau, \\ \widehat{m}_{i+n}^{\rightarrow} \cup \widehat{m}_{i+n}^{\leftarrow}, & \text{otherwise.} \end{cases}
\tag{5}
$$

If SAM2 (Ravi et al., 2024) fails to propagate a mask from one side, we fall back to the valid direction:

$$
m_{i+n} = \begin{cases} \widehat{m}_{i+n}^{\leftarrow}, & \text{if } \widehat{m}_{i+n}^{\rightarrow} = \varnothing, \\ \widehat{m}_{i+n}^{\rightarrow}, & \text{otherwise.} \end{cases}
\tag{6}
$$

Our bidirectional fusion produces temporally consistent masks and only adds a minimal runtime overhead(Appendix A.9). By leveraging both past and future context to reconcile forward- and backward-propagated masks, the fusion step not only improves robustness in dynamic scenes but also helps to correct occasional pointing errors from Stage 1—recovering from mislocalized points—and thereby yields higher-quality segmentation masks. The proposed bidirectional fusion pipeline generalizes effectively to other segmentation architectures. Comprehensive results across different segmentation backbones are provided in Appendix A.5.

## 5 VIDEOMOLMO DATASET

**Training Data:** Our dataset is designed to train the model's spatio-temporal pointing capabilities. It contains 72k video–caption pairs with annotated object points, sourced from diverse benchmarks spanning video object segmentation, tracking, and referring expression tasks, including Refer-YouTube-VOS (Seo et al., 2020b), Refer-DAVIS (Perazzi et al., 2016), MeViS (Ding et al., 2023), GroOT (Nguyen et al., 2023), LaSOT (Fan et al., 2019), ViCaS-LGVIS (Athar et al., 2024), and Reason-VOS (Yan et al., 2024). To create fine-grained data grounded in point coordinates, we develop a semi-automated annotation pipeline (see Fig. 3) that ensures both high-quality and scalable annotations. Each of the above-mentioned datasets features video-mask-expression triplets

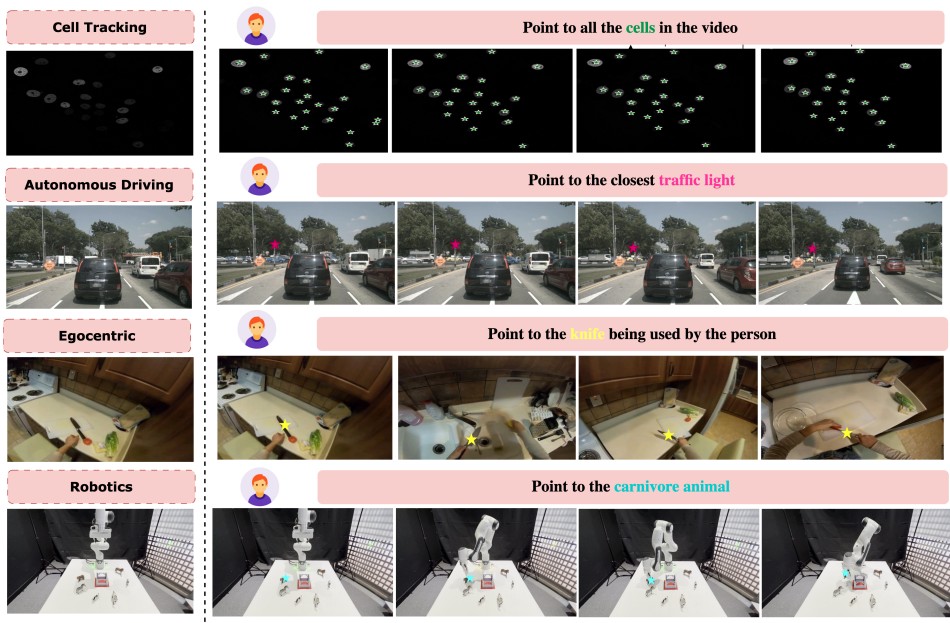

Figure 4: VIDEOMOLMO demonstrates robust generalization and fine-grained spatio-temporal grounding across diverse out-of-distribution scenarios from our proposed benchmark, for instance, correctly pointing to traffic lights (2$^{nd}$ row) in challenging driving scenes despite never encountering such scenarios during training. (Please refer to Appendix A.11 for additional qualitative results.)

$(V, M, E)$ such that $V \in \mathbb{R}^{|\mathcal{T}| \times H \times W \times C}$, $M \in \{0, 1\}^{|\mathcal{T}| \times |\mathcal{O}| \times H \times W}$ with $|\mathcal{O}|$ denoting the number of unique annotated objects in the frame $\mathcal{T}_i$. For each object $\mathcal{O}_j \in \mathcal{O}$ in frame $\mathcal{T}_i \in \mathbb{R}^{H \times W \times C}$, with a binary mask $m_j \in \{0, 1\}^{H \times W}$, the goal is to extract a single highly representative point co-ordinate for the object. We sample $k$ candidate points within the mask, assigning each point $(x, y)$ a sampling probability proportional to its Euclidean distance to the nearest boundary pixel of the mask $m_j$, i.e.,

$$P(x, y) \propto \min_{(x', y') \in \partial m_j} \|(x, y) - (x', y')\|_2 \tag{7}$$

where $\partial m_j$ denotes the set of boundary pixels of the mask. For each sampled point, we use SAM2 (Ravi et al., 2024) to predict a segmentation mask. We then compute the Intersection-over-Union (IoU) between each predicted mask and the corresponding ground truth mask $m_j$. The point coordinate whose predicted mask achieves the highest IoU is selected as the representative ground truth point for the object:

$$p^* = \arg\max_{(x, y)} \text{IOU}\left(\text{SAM2}(x, y), \, m_j\right) \tag{8}$$

where $\text{SAM2}(x, y)$ denotes the predicted mask obtained using point $(x, y)$ as a prompt to SAM2 (Ravi et al., 2024).

**VPoS-Bench:** To evaluate the generalization capabilities of VIDEOMOLMO, we introduce Video Pointing and Segmentation (VPoS-Bench) benchmark, a curated benchmark test set comprising 100 video–caption pairs and 1k manually annotated object points. For mask-based evaluations, we employ SAM2 (Ravi et al., 2024) to convert point annotations into segmentation masks. The benchmark covers diverse real-world scenarios drawn from both open datasets (Caesar et al., 2020; Lin et al., 2024; Grauman et al., 2022; Singh et al., 2024; Maška et al., 2023) and internal collections, spanning five categories: Cell Tracking, Egocentric Videos, Autonomous Driving, Video-GUI, and Robotics. In addition, VPoS-Bench includes a dedicated counting task derived from the (Kay et al., 2017) dataset. Further details about the benchmark are provided in Appendix A.3.

## 6 EXPERIMENTS

**Implementation details.** VIDEOMOLMO follows the architecture of Molmo (Deitke et al., 2024). For the image encoder, we use a pretrained CLIP ViT-L/14 (336 × 336) (Radford et al., 2021) model.

Table 1: Performance of various models on five subtasks of VPoS-Bench (Ego4D, Robotics, Autonomous, Cells, VideoGUI)

| Model | Ego4D | | | Robotics | | | Autonomous | | | Cells | | | VideoGUI | | |
|---|---|---|---|---|---|---|---|---|---|---|---|---|---|---|---|
| | $\mathcal{J}$ | $\mathcal{F}$ | $\mathcal{J\&F}$ | $\mathcal{J}$ | $\mathcal{F}$ | $\mathcal{J\&F}$ | $\mathcal{J}$ | $\mathcal{F}$ | $\mathcal{J\&F}$ | $\mathcal{J}$ | $\mathcal{F}$ | $\mathcal{J\&F}$ | $\mathcal{J}$ | $\mathcal{F}$ | $\mathcal{J\&F}$ |
| VideoLISA (Bai et al., 2024) | 47.1 | 41.1 | 44.2 | 3.9 | 2.0 | 2.9 | 34.7 | 22.0 | 28.4 | 18.9 | 3.3 | 11.1 | 65.3 | 39.4 | 52.4 |
| VideoGLaMM (Munasinghe et al., 2024) | 47.2 | 40.4 | 43.8 | 15.3 | 10.3 | 12.8 | 31.4 | 17.7 | 24.8 | 11.8 | 7.8 | 9.8 | 58.7 | 32.5 | 45.6 |
| Molmo (Deitke et al., 2024) +SAM2 (Ravi et al., 2024) | 50.6 | 50.1 | 50.4 | 27.8 | 25.6 | 26.7 | 49.5 | 47.5 | 48.5 | 20.8 | 7.6 | 14.2 | 60.2 | 55.9 | 58.0 |
| VIDEOMOLMO | **55.5** | **54.3** | **54.9** | **29.1** | **26.2** | **27.6** | **57.1** | **57.7** | **57.4** | **25.4** | **13.1** | **19.2** | **68.9** | **62.4** | **65.7** |

Table 2: Performance comparison on Refer-DAVIS-17, Refer-YouTube-VOS, and MeViS benchmarks. VideoMolmo consistently improves referring video object segmentation across datasets. (" - " indicates missing results in prior literature)

| Model | Refer-DAVIS-17 | | | Refer-YouTube-VOS | | | MeViS | | |
|---|---|---|---|---|---|---|---|---|---|
| | $\mathcal{J}$ | $\mathcal{F}$ | $\mathcal{J\&F}$ | $\mathcal{J}$ | $\mathcal{F}$ | $\mathcal{J\&F}$ | $\mathcal{J}$ | $\mathcal{F}$ | $\mathcal{J\&F}$ |
| LISA-7B (Lai et al., 2023) | 61.9 | 54.9 | 58.4 | 50.6 | 49.7 | 50.2 | – | – | – |
| LISA-13B (Lai et al., 2023) | 64.6 | 56.8 | 60.7 | 53.0 | 52.1 | 52.6 | – | – | – |
| TrackGPT-7B (Zhu et al., 2023b) | 67.0 | 59.4 | 63.2 | 57.4 | 55.3 | 56.4 | – | – | – |
| TrackGPT-13B (Zhu et al., 2023b) | 70.4 | 62.7 | 66.5 | 60.8 | 58.1 | 59.5 | – | – | – |
| VideoLISA (Bai et al., 2024) | 72.7 | 64.9 | 68.8 | **65.7** | 61.7 | 63.7 | 41.3 | 47.6 | 44.4 |
| VideoGLaMM (Munasinghe et al., 2024) | **73.3** | 65.6 | 69.5 | 65.4 | 68.2 | 66.8 | 42.1 | 48.2 | 45.2 |
| Molmo (Deitke et al., 2024)+SAM2 (Ravi et al., 2024) | 65.3 | 72.2 | 68.8 | 61.0 | 66.2 | 63.6 | 44.4 | 49.4 | 46.9 |
| VIDEOMOLMO | 71.3 | **73.6** | **72.5** | 65.6 | **69.1** | **67.3** | **51.2** | **56.6** | **53.9** |

Our proposed Temporal Module is initialized from scratch. Our choice of LLM is the pretrained Qwen2-7B (Yang et al., 2024). We train the model on 8 NVIDIA A100 80GB GPUs. Learning rate of $1e^{-5}$ is used for the LLM, and $5e^{-6}$ for the vision encoder, visual projector and temporal module. We adopt a batch size of 1 with 256 gradient accumulation steps, and use AdamW optimizer (Loshchilov & Hutter, 2019) following the fine-tuning recipe from (Deitke et al., 2024). During inference, we use sampling rate of 5 for segmentation tasks and 10 for point grounding and counting tasks. Please refer Appendix A.1.1 and A.1.2 for more training and inference details.

**Tasks.** We evaluate VIDEOMOLMO on four challenging tasks: (1) point grounding, (2) counting, (3) referring segmentation, and (4) reasoning video object segmentation. For point grounding, we report performance on our proposed VPoS-Bench. For the counting task, we utilize videos from the Kinetics dataset (Kay et al., 2017), where object counts range from $2 - 13$. For referring video segmentation, we use MeViS validation set (Ding et al., 2023), Refer-DAVIS-17 (Khoreva et al., 2019) and Refer-YouTube-VOS (Seo et al., 2020a) datasets. Finally, for reasoning segmentation, we evaluate our model on the ReasonVOS dataset (Bai et al., 2024).

**Evaluation metrics.** For the point grounding task, we follow the evaluation protocol of Molmo (Deitke et al., 2024) and report *Precision*: the fraction of predicted points that fall inside the corresponding ground-truth mask, *Recall*: the fraction of ground-truth objects that are successfully localized by at least one predicted point, and *F1 score*: the harmonic mean of Precision and Recall, balancing both metrics. For mask-based evaluations, we use Region Jaccard ($\mathcal{J}$): intersection-over-union (IoU) between predicted and ground-truth masks, Boundary F-measure ($\mathcal{F}$): F1 score computed on the precision and recall of boundary pixels between predicted and ground-truth masks, and and their average $\mathcal{J\&F}$, widely used as a comprehensive mask similarity measure. For the counting task, we report *Exact Matching Accuracy* (EMA): the percentage of predictions where the predicted count exactly matches the ground-truth count, and *Mean Absolute Error* (MAE): the average absolute difference between predicted and ground-truth counts.

**Baselines.** For point grounding, we compare VIDEOMOLMO with three strong baselines: VideoLISA(Bai et al., 2024), VideoGLaMM (Munasinghe et al., 2024), and Molmo+SAM2. To adapt Molmo for videos, we augment it with SAM2. For referring segmentation, we evaluate against VideoLISA, VideoGLaMM, and prior baselines. For counting, we compare VIDEOMOLMO with both closed-source (GPT-5 (OpenAI, 2025)) and open-source models (Deitke et al., 2024; Bai et al., 2025; Team et al., 2024)). For further experimentation details, please refer to Appendix A.2.

## 6.1 MAIN EXPERIMENTATION RESULTS

**Point Grounding.** The point grounding task focuses on accurately identifying the spatial coordinates of a queried object within video frames. As depicted in Fig. 5, VIDEOMOLMO demonstrates superior performance in localizing target points, as evidenced by its significantly higher Precision, Recall, and F1 scores compared to Molmo. This performance gap can be attributed to Molmo's training on static frames, which limits its ability to handle temporal variations. In dynamic video inputs,

Table 3: Performance comparison of Video-Molmo on the ReasonVOS benchmark.

| Model | $\mathcal{J}$ | $\mathcal{F}$ | $\mathcal{J\&F}$ |
|---|---|---|---|
| LISA (Lai et al., 2023) | 33.1 | 29.1 | 31.1 |
| VideoLISA (Bai et al., 2024) | **49.9** | 45.1 | 47.5 |
| VideoGLaMM (Munasinghe et al., 2024) | 40.5 | 27.2 | 33.9 |
| Molmo (Deitke et al., 2024) + SAM2 (Ravi et al., 2024) | 43.5 | 47.8 | 45.7 |
| VIDEOMOLMO | 48.7 | **53.4** | **51.1** |

Table 4: Performance comparison of Video-Molmo on the counting benchmark. (($\downarrow$ lower is better, $\uparrow$ higher is better.)

| Model | MAE $\downarrow$ | EMA $\uparrow$ |
|---|---|---|
| GPT-5 (OpenAI, 2025) | 0.76 | 60.0 |
| Gemma3-12B (Team et al., 2024) | 0.96 | 43.3 |
| Qwen2.5-VL-7B (Bai et al., 2025) | 0.83 | 50.0 |
| Molmo (Deitke et al., 2024) | 1.21 | 49.3 |
| VIDEOMOLMO | **0.72** | **73.3** |

where object presence and position may vary across frames, Molmo struggles, whereas VIDEO-MOLMO effectively addresses this challenge by leveraging temporal context. Furthermore, VIDEO-MOLMO outperforms all baseline models across each subtask from VPoS-Bench, as evident from higher $\mathcal{J}$, $\mathcal{F}$, and the combined $\mathcal{J\&F}$ metric (Table 1). Qualitative results in Fig.4 further validate the robustness of VIDEOMOLMO, showcasing its ability to accurately localize objects across diverse and out-of-distribution scenarios.

**Object Counting.** VPoS-Bench introduces a dedicated object counting task, essential for many real-world video understanding applications. Object counts range from $2 - 13$, requiring enhanced temporal and spatial reasoning. We compare VIDEOMOLMO against both open-source and proprietary models (Table 4). VIDEOMOLMO achieves state-of-the-art performance, outperforming all baselines in MAE and EMA, and even surpasses advanced proprietary models like GPT-5 (OpenAI, 2025), highlighting its strength as a specialized counting model. This success stems from training VideoMolmo on our large-scale comprehensive dataset with multiple objects per video, enabling fine-grained multi-object understanding. We also evaluate VIDEOMOLMO on videos with a large number of objects ($> 30$), with results in Appendix A.10.

**Referring Segmentation.** For referring video segmentation, the goal is to localize specific object instances in a video based on a given phrase. Table 2 presents results across three standard datasets. On the MeViS benchmark, which involves motion-guided segmentation with multiple objects, VIDEOMOLMO outperforms all baselines by a notable margin, demonstrating its effectiveness in grounding complex, multi-object scenes. This advantage stems in part from the simplicity and efficiency of VIDEOMOLMO's point-based supervision as reflected in its superior $\mathcal{J}$, $\mathcal{F}$, and $\mathcal{J\&F}$ scores, which contrasts with recent methods like VideoGLaMM (Munasinghe et al., 2024) and VideoLISA (Bai et al., 2024) that rely on dense, pixel-level mask prediction where precise delineation between objects becomes challenging (Fig. 1). VIDEOMOLMO also achieves superior performance on Refer-DAVIS-17 and Refer-YouTube-VOS. Notably, VideoGLaMM performs competitively on Refer-YouTube-VOS which features fast-moving objects, benefiting from its dual encoder architecture that integrates spatial and temporal features. Despite relying on a single encoder with point-based supervision, VIDEOMOLMO surpasses strong baselines, aided by its temporal module, novel post-processing, and point-grounding paradigm.

**Reasoning Video Object Segmentation.** Table 3 presents a comparative analysis of VIDEO-MOLMO against existing approaches on the ReasonVOS benchmark, which emphasizes complex reasoning, temporal comprehension, and consistency across frames, making it particularly challenging. Prior methods perform noticeably worse, largely due to their limited temporal and fine-grained reasoning capabilities. While VideoLISA incorporates spatio-temporal cues, it still falls short of VIDEOMOLMO. This performance gap highlights VIDEOMOLMO's architectural strengths, specifically its dedicated temporal module providing rich spatio-temporal contextual understanding.

## 6.2 ABLATIONS AND ANALYSIS

**Effect of Temporal Module.** We conduct an ablation study to evaluate the effectiveness of different temporal module variants on the Refer-DAVIS benchmark (Table 6). Using a single frame or simple feature fusion methods such as addition or token-space concatenation yields relatively lower performance compared to our proposed cross-attention-based temporal module as it enables dynamic and selective integration of relevant features across frames, allowing the model to focus on temporally coherent and semantically meaningful cues critical for accurate grounding.

**Ablation on Temporal Mask Fusion:** To enable efficient and temporally consistent segmentation, we evaluate various strategies for combining masks propagated from the sampled frames. As shown in Table 5, naive strategies like preferring left/right predictions or computing mask intersections result in suboptimal performance, either due to loss of temporal context or overly conservative fusion. Our proposed bidirectional fusion strategy outperforms all baselines by adaptively reconciling

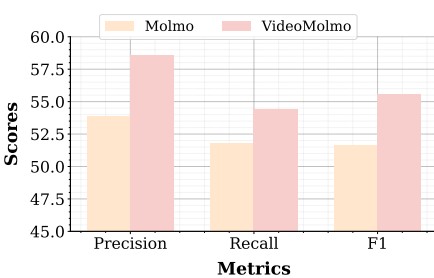

Figure 5: Performance comparison of Video-Molmo on point grounding.

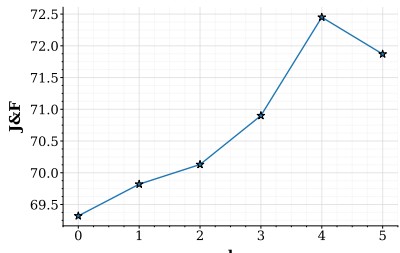

Figure 6: Effect of temporal module context length on segmentation accuracy in Refer-DAVIS benchmark.

Table 5: Effect of different temporal mask fusion strategies on Refer-DAVIS dataset.

| Strategy | $\mathcal{J}$ | $\mathcal{F}$ | $\mathcal{J}\&\mathcal{F}$ |
|---|---|---|---|
| Prefer left | 67.31 | 73.38 | 70.34 |
| Prefer right | 67.05 | 71.69 | 69.37 |
| Intersection | 60.20 | 63.58 | 61.89 |
| Larger mask | 70.40 | 72.91 | 71.65 |
| Smaller mask | 64.10 | 67.18 | 65.64 |
| VIDEOMOLMO | **71.27** | **73.63** | **72.45** |

Table 6: Ablation on different variants of the temporal module on the Refer-DAVIS dataset.

| Variant | $\mathcal{J}$ | $\mathcal{F}$ | $\mathcal{J}\&\mathcal{F}$ |
|---|---|---|---|
| Single frame | 66.04 | 72.60 | 69.32 |
| Addition | 66.13 | 72.97 | 69.55 |
| Concatenation | 65.34 | 72.06 | 68.71 |
| VIDEOMOLMO | **71.27** | **73.63** | **72.45** |

forward and backward propagated masks based on their agreement (IoU). Our fallback mechanism ensures robustness against failure cases where one of the propagated masks is missing. This approach achieves a significant improvement in $\mathcal{J}\&\mathcal{F}$ score of 72.45, demonstrating its effectiveness.

**Effect of context-length in temporal module:** To analyze the effect of context length in the temporal module, we evaluate VIDEOMOLMO on the Refer-DAVIS benchmark (Fig. 6). We observe that there is a consistent increase in $J\&F$ as the context length increases from $1 \rightarrow 4$, indicating that incorporating more temporal information enhances the model's spatio-temporal reasoning. However, there is a slight drop in accuracy at $l = 5$, suggesting that adding more frames may introduce redundancy or noise rather than useful context.

**Effect of multiple points.** We evaluate the impact of increasing the number of predicted points, since SAM2 (Ravi et al., 2024) naturally supports multi-point prompting. As shown in Table 13, additional points provide only marginal gains in $J\&F$, suggesting that the single point predicted by VIDEOMOLMO already offers a strong representation for our bidirectional temporal mask fusion. However, in specific cases such as thin and elongated objects, multiple points can be beneficial (see Tables 12 and 11). We further explore prompting SAM2 with negative points (Table 14). Detailed ablations are provided in Appendix A.4, with additional results in Appendix A.1. Finally, we analyze our choice of point→mask strategy over mask→point approach in Appendix A.8.

## 7 CONCLUSION

We present VIDEOMOLMO, a Video-LMM for fine-grained spatio-temporal pointing conditioned on textual queries. It leverages a temporal module that incorporates temporal cues from preceding frames and a novel bidirectional post-processing strategy for robust mask prediction. To enable training, we curate a large-scale spatio-temporal pointing dataset using a semi-automatic annotation pipeline. VIDEOMOLMO shows strong generalization and consistently outperforms state-of-the-art models across diverse and out-of-distribution tasks, including point grounding, object counting, referring segmentation, and reasoning segmentation.

**Limitations and Future Work.** VIDEOMOLMO demonstrates strong spatio-temporal grounding performance, excelling in fine-grained localization without requiring explicit pixel-level mask supervision. However, its performance might degrade on videos with fast-moving objects, due to single-frame processing during training—an efficiency-driven design choice constrained by both architectural and computational limits. In addition, the quality of final mask predictions remains bounded by the performance of SAM2 (Ravi et al., 2024). Finally, although our VPoS-Bench covers diverse domains, its relatively modest scale may limit the assessment of generalization to more complex real-world scenarios. Future work may explore joint multi-frame training with improved sampling strategies, and expansion of benchmarks to broader, large-scale settings.

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

# A APPENDIX

## A.1 ADDITIONAL ABLATIONS

### A.1.1 TRAINING ABLATIONS

In addition to the ablation studies presented in the main paper, we conduct further investigations into the impact of language backbone choice, parameter tuning, and numerical precision on the Ref-DAVIS dataset, as summarized in Table 7.

In the first row, we assess the effect of replacing the Qwen2-7B language model with Olmo-7B. The resulting drop in $\mathcal{J}\&\mathcal{F}$ score highlights Qwen2-7B's superior grounding capabilities which is consistent with the observational findings reported in (Deitke et al., 2024). This emphasizes the importance of selecting an Qwen2-7B LLM with strong multimodal alignment for visual grounding tasks.

Table 7: Performance comparison of different VIDEOMOLMO variants on Refer-DAVIS benchmark.

| Variant | $J$ | $F$ | $J\&F$ |
|---|---|---|---|
| VIDEOMOLMO-O-7B | 66.25 | 72.93 | 69.59 |
| VIDEOMOLMO (LoRA) | 67.82 | 74.72 | 71.27 |
| VIDEOMOLMO (16bit) | 67.74 | 74.72 | 71.25 |
| VIDEOMOLMO | **71.27** | **73.63** | **72.45** |

Next, we investigate the impact of end-to-end training. In the second row, we freeze the video encoder and train only the LLM's projection layers by integrating LoRA (Hu et al., 2022) adapters. This lightweight training strategy significantly underperforms compared to the fully fine-tuned model (last row), validating our hypothesis that joint optimization of all components is essential for capturing the temporal nuances required for precise point grounding.

Finally, we examine the effect of training precision. In the third row, we use 16-bit floating point precision which is commonly adopted to save memory and accelerate training. However, we find that this leads to a notable degradation in performance. In contrast, training with full 32-bit precision (last row) enhances the model's capacity to learn fine-grained spatial and temporal cues, consistent with prior observations in (Deitke et al., 2024).

Together, these ablations underline the significance of careful backbone selection, full end-to-end optimization, and high-precision training for achieving robust and fine-grained visual grounding in VIDEOMOLMO.

### A.1.2 INFERENCE ABLATIONS

**Sampling rate $k$:** As described in the main paper, we adopt a frame sampling rate of $k = |\mathcal{T}|$ for the Molmo+SAM2 baseline during inference, which means that we take the first frame prediction and use SAM2 to propagate the mask across all the frames. This choice is motivated by performance on the Refer-DAVIS-17 dataset, where Molmo+SAM2 achieves its highest $\mathcal{J}\&\mathcal{F}$ score of 67.69 at this value. However, our analysis in Table 8 reveals that the optimal sampling rate is not universal, it varies across datasets. To ensure a fair and competitive comparison with our proposed VIDEOMOLMO, we conduct additional ablations on the sampling rate $k$ for Molmo+SAM2 across the Refer-YouTube-VOS and MeViS datasets. We find that a lower sampling rate of $k = 3$ yields the best performance on Refer-YouTube-VOS, while $k = 1$ proves optimal on MeViS. Despite this tuning, VIDEOMOLMO consistently outperforms Molmo+SAM2 under each dataset's optimal configuration. Interestingly, across all three datasets, we observe a consistent decline in performance as the sampling rate increases. This is particularly evident at $k = 30$, where the baseline performance starts dropping. These findings further highlight the robustness of VIDEOMOLMO in leveraging temporal context, even when competing baselines are tuned to their best-performing configurations.

We further ablate the effect of sampling rate on our proposed VIDEOMOLMO. While our main results on the Refer-YouTube-VOS benchmark in the main paper are reported using a sampling rate of $k = 5$, we acknowledge that this choice, although consistent with the baseline configuration, may

Table 8: Effect of sampling rate $k$ on Refer-DAVIS-17, Refer-Youtube-VOS, and MeViS benchmarks using Molmo + SAM2

| $k$ | Refer-DAVIS-17 | | | Refer-Youtube-VOS | | | MeViS | | |
|---|---|---|---|---|---|---|---|---|---|
| | $\mathcal{J}$ | $\mathcal{F}$ | $\mathcal{J}\&\mathcal{F}$ | $\mathcal{J}$ | $\mathcal{F}$ | $\mathcal{J}\&\mathcal{F}$ | $\mathcal{J}$ | $\mathcal{F}$ | $\mathcal{J}\&\mathcal{F}$ |
| 1 | 58.32 | 64.60 | 61.46 | 60.08 | 64.69 | 62.38 | 45.53 | 51.06 | 48.30 |
| 3 | 58.77 | 63.85 | 61.31 | 60.11 | 65.04 | 62.58 | 45.58 | 50.97 | 48.27 |
| 10 | 59.61 | 64.77 | 62.19 | 60.24 | 64.88 | 62.56 | 45.63 | 50.93 | 48.28 |
| 30 | 63.31 | 69.31 | 66.31 | 59.48 | 64.02 | 61.75 | 45.51 | 50.65 | 48.08 |
| $|\mathcal{T}|$ | 65.29 | 70.09 | 67.69 | 59.48 | 64.07 | 61.78 | 44.37 | 49.37 | 46.87 |

Table 9: Effect of varying threshold $\tau$ on the performance of VIDEOMOLMO evaluated on the Refer-DAVIS benchmark.

| $\tau$ | $\mathcal{J}$ | $\mathcal{F}$ | $\mathcal{J}\&\mathcal{F}$ |
|---|---|---|---|
| 0 | 67.31 | 73.38 | 70.34 |
| 0.3 | 69.05 | 75.51 | 72.28 |
| 0.5 | 68.90 | 75.26 | 72.08 |
| 0.9 | 68.85 | 75.24 | 72.05 |
| VIDEOMOLMO ($\tau = 0.7$) | **71.27** | **73.63** | **72.45** |

Table 10: Effect of sampling rate $k$ on the performance of VIDEOMOLMO evaluated on the Refer-YouTube-VOS benchmark.

| $k$ | $\mathcal{J}$ | $\mathcal{F}$ | $\mathcal{J}\&\mathcal{F}$ |
|---|---|---|---|
| 3 | 64.39 | 67.68 | 66.03 |
| 10 | 65.69 | 69.39 | 67.54 |
| 15 | 66.26 | 69.95 | 68.11 |
| 20 | 66.34 | 69.93 | 68.14 |
| 30 | 65.80 | 69.32 | 67.56 |
| VIDEOMOLMO ($k = 5$) | **65.55** | **69.11** | **67.33** |

not be optimal for our method. As seen from the Table 10, VIDEOMOLMO benefits from careful selection of sampling rate as we observe that a sampling rate $k = 20$ yields the highest $\mathcal{J}\&\mathcal{F}$ score of 68.14, compared to 67.33 with $k = 5$.

**Threshold $\tau$:** Our proposed post-processing strategy, *Bidirectional Temporal Mask Fusion*, enhances model performance by combining masks propagated from both right and left directions to achieve a robust temporal consensus. As described in Section 4.2 of the main paper, the fusion process is governed by a threshold hyperparameter $\tau$, which determines how agreement between the two masks is evaluated. Specifically, when the Intersection-over-Union (IoU) between the forward and backward masks exceeds $\tau$, their intersection is used as the final mask, enforcing stricter agreement. Conversely, if the IoU falls below $\tau$, their union is taken, promoting flexibility in ambiguous regions. This mechanism balances precision and recall based on temporal consistency. We ablate different values of $\tau$ in Table 9 to identify the most effective setting. The results indicate that $\tau = 0.7$ yields the best overall performance. However, the differences across values are relatively minor, underscoring the high quality and stability of the point predictions generated by VIDEOMOLMO. This consistency highlights the robustness of our model in temporal point grounding, even under varying post-processing thresholds.

**Effect of Post-processing on baselines:** To assess the generalizability and effectiveness of our proposed *Bidirectional Temporal Mask Fusion*, we integrate it with the Molmo+SAM2 baseline, resulting in an enhanced variant denoted as Molmo$^\dagger$+SAM2. As illustrated in Fig. 7, this integration consistently improves performance across all three datasets in terms of $\mathcal{J}\&\mathcal{F}$ score. These results demonstrate that our post-processing strategy not only strengthens our own model but also benefits existing methods. The modular, plug-and-play nature makes it a valuable addition to any video grounding pipeline, improving temporal consistency and overall segmentation quality with minimal integration effort.

## A.2 ADDITIONAL EXPERIMENTATION DETAILS

VIDEOMOLMO follows the architecture of Molmo (Deitke et al., 2024). For the image encoder, we use a pretrained CLIP ViT-L/14 ($336 \times 336$) (Radford et al., 2021) model. We initialize the temporal module with Xavier Normalized weights for stable training. Our choice of LLM is the pretrained Qwen2 7B (Yang et al., 2024). We train the model on 8 NVIDIA A100 80GB GPUs. Learning rate of $1e^{-5}$ is used for the LLM, and $5e^{-6}$ for the vision encoder, visual projector and temporal module. We adopt a batch size of 1 with 256 gradient accumulation steps, and use AdamW optimizer with $\beta = (0.9, 0.95)$ and $\epsilon = 1e^{-6}$, . We train VIDEOMOLMO for 4 epochs on 8 NVIDIA A100 GPUs (80 GB each), consuming roughly 1,000 GPU-hours in total. Training runs in full 32-bit

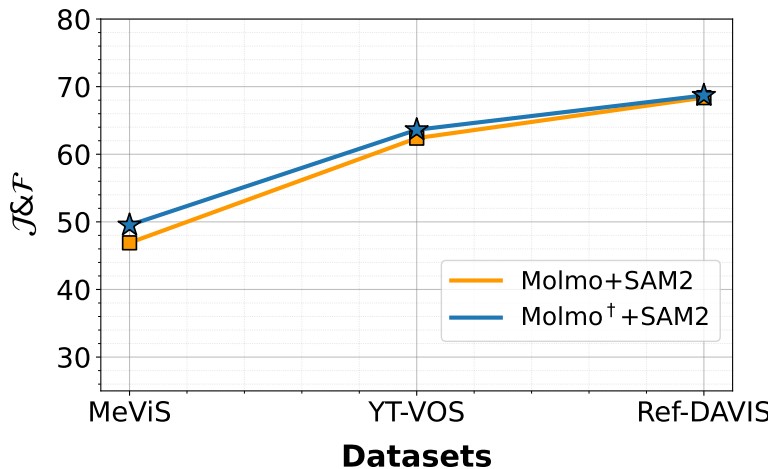

Figure 7: Effect of *Bidirectional temporal mask fusion* on Molmo+SAM2 baseline.

precision with a 10-step linear warmup, after which we follow a cosine learning-rate schedule; we also clip gradients to a maximum norm of $1.0$ to guard against unstable updates. For all inference and reported results, we use 4-bit precision.

### A.3 VPoS BENCH

As mentioned in the main paper, we introduce Video Pointing and Segmentation (VPoS-Bench), a curated benchmark test set comprising of 100 video-caption pairs and $1k$ manually annotated object points. To obtain mask annotations for evaluation, we use SAM2 to convert these point annotations into segmentation masks. For mask-based evaluations, we employ the SAM2 model to convert these point annotations into segmentation masks. The test benchmark encompasses diverse real-world scenarios, sourced from both open datasets (Caesar et al., 2020; Lin et al., 2024; Grauman et al., 2022) and internal collections, spanning five categories: Cell Tracking, Egocentric Videos, Autonomous Driving, Video-GUI, and Robotics. Our benchmark also consists of a dedicated counting task sourced from (Kay et al., 2017) dataset. Below, we present details about each subset in VPoS-Bench.

*1) Cell Tracking:* Features internally sourced 12 microscopic videos with dynamic cellular structures, where precise localization of individual cells is essential for tasks like tracking cell division or counting. These videos are partially sourced from (Maška et al., 2023) and the remaining are requested internally.

*2) Egocentric Videos:* Comprises 18 first-person videos capturing daily human-object interactions, enabling the assessment of grounded pointing in scenarios such as object manipulation and activity recognition. The egocentric videos in our test benchmark are derived from (Grauman et al., 2022) dataset.

*3) Autonomous Driving:* Includes 13 urban driving scenes from nuScenes's dataset (Caesar et al., 2020), with complex environments, requiring accurate identification of specific road elements (e.g., traffic signals) to support navigation and safety systems.

*4) Video-GUI:* Consists of 13 screen recordings from software applications, focusing on tasks like identifying and interacting with user interface elements based on textual instructions. The VideoGUI videos are sampled from VideoGUI dataset (Lin et al., 2024).

*5) Robotics:* Encompasses 14 videos of robotic operations, emphasizing the need for precise object localization to execute commands such as "pick up the red block" or "press the top button." Few of the robotic videos in our benchmark are sourced from (Singh et al., 2024), and the remaining are sourced internally.

## A.4 LIMITATIONS OF SINGLE-POINT PREDICTION

VIDEOMOLMO can be run once to produce a single point or repeatedly to generate $N$ points per object, and it can also predict multiple points in one forward pass when conditioned on the prompt. To quantify the benefit of multiple points, we performed detailed ablations on the referring segmentation task.

While we observed no significant change in overall $\mathcal{J}\&\mathcal{F}$ on Refer-DAVIS2017 likely because most objects in this dataset are not highly elongated, there are clear **failure cases** where relying on a single point leads SAM2 astray.

**Analysis of failure cases.** We present qualitative failure cases in Fig. 12 . Below, we illustrate two examples where providing multiple points significantly improves the mask quality generated by SAM2.

**Bag vs. Person:** When asked to segment a bag, SAM2 with a single point often includes part of the person or misses sections of the bag. Providing multiple points on the bag helps SAM2 focus and recover a more complete mask (Table 11

Table 11: Effect of single vs. multiple points on segmenting a bag.

| Method | $\mathcal{J}$ | $\mathcal{F}$ | $\mathcal{J}\&\mathcal{F}$ |
|---|---|---|---|
| Single Point | 28.5 | 36.3 | 32.4 |
| Multiple Points | 38.6 | 43.4 | 41.0 |

**Paraglider Lines:** Elongated, thin structures such as paraglider lines are challenging to segment with a single point, whereas distributing multiple points along the lines enables a more complete reconstruction of the object (Table 12).

Table 12: Effect of single vs. multiple points on segmenting paraglider lines.

| Method | $\mathcal{J}$ | $\mathcal{F}$ | $\mathcal{J}\&\mathcal{F}$ |
|---|---|---|---|
| Single Point | 2.5 | 5.1 | 3.8 |
| Multiple Points | 47.8 | 67.2 | 57.5 |

**Ablation on number of points.** We further vary the number of points per object on Refer-DAVIS2017. As shown in Table 13, the $\mathcal{J}\&\mathcal{F}$ score quickly plateaus, suggesting that most objects do not require more than one point for SAM2 to succeed.

Table 13: Ablation of number of points per object on Refer-DAVIS2017.

| # Points | $\mathcal{J}\&\mathcal{F}$ |
|---|---|
| 1 | 72.5 |
| 2 | 72.7 |
| 3 | 72.8 |
| 4 | 72.8 |

We observe that general performance on standard benchmarks remains high with a single point. However, edge cases involving elongated or intricate boundaries benefit greatly from adaptively selecting multiple points, as shown above. Incorporating an $N$-point strategy provides a simple and effective remedy for such extreme examples, without hurting overall accuracy, though at the cost of increased complexity.

We also evaluate prompting SAM2 (Ravi et al., 2024) with negative points, where VIDEOMOLMO is asked to indicate background regions instead of the target object. This yields only a marginal gain of 0.2 pp in accuracy on the Ref-DAVIS benchmark (Perazzi et al., 2016)(Table 14), suggesting that mask quality is primarily bounded by the underlying performance of SAM2. We expect these results to improve with stronger segmentation backbones.

Table 14: Ablation of negative points as prompts to SAM2 in Refer-DAVIS2017.

| Method | $\mathcal{J}\&\mathcal{F}$ |
|---|---|
| Single Positive | 72.5 |
| Positive and Negative | 72.6 |
| Multiple Points | 72.8 |

## A.5 Generalizability and Biasness of Using SAM2

To assess the generalizability of using SAM2 (Ravi et al., 2024) for identifying the most representative point during annotation, we conducted additional experiments by integrating VIDEOMOLMO with two alternative segmentation models: (Zhao et al., 2023b): A YOLO-based model with a fundamentally different architecture from SAM2 (Ravi et al., 2024) and MobileSAM (Zhang et al., 2023a): A lightweight, distilled variant of SAM (Kirillov et al., 2023) that uses a ViT-based encoder.

Our choice of SAM2 (Ravi et al., 2024) is strategic. As the current state-of-the-art open-vocabulary segmentation model, SAM2 (Ravi et al., 2024) provides a robust foundation for selecting high-quality candidate points across diverse object categories and challenging scenarios. Its strong handling of edge cases and domain variability ensures the creation of reliable annotations that generalize well. In contrast, weaker segmentation models may introduce systematic biases and degrade the quality of learned representations.

Table 15: Performance comparison of VIDEOMOLMO when integrated with different segmentation models.

| Segmentation Model | $\mathcal{J}\&\mathcal{F}$ |
|---|---|
| FastSAM | 59.7 |
| MobileSAM | 62.3 |
| SAM2 (VIDEOMOLMO) | **72.5** |

Despite lacking temporal modeling and differing in architecture, both FastSAM (Zhao et al., 2023b) and MobileSAM (Zhang et al., 2023a) achieve reasonably strong results (59.7 and 62.3), retaining 82–86% of SAM2's performance (Table 15). This indicates that our point annotations encode **fundamental spatial localization cues** that generalize across segmentation paradigms. We attribute the performance gap of $10-13$ points primarily to the absence of temporal consistency, rather than poor point quality. Importantly, SAM2 (Ravi et al., 2024) remains the only video-native foundation model that supports point-based prompts, making it the most practical and capable option for building training data with temporal consistency. Its memory mechanisms and temporal propagation provide capabilities that FastSAM (Zhao et al., 2023b)and MobileSAM (Zhang et al., 2023a) lack.

Finally, our comprehensive evaluation on **VPoS-Bench** confirms that this design yields strong cross-domain generalization and robustness. Across five diverse out-of-distribution domains— Cell Tracking, Egocentric Vision, Autonomous Driving, Video-GUI Interaction, and Robotics— VIDEOMOLMO consistently outperforms baselines with an average improvement of 5.4 percentage points. This result underscores that our learned pointing representations capture core object localization principles that generalize beyond the characteristics or potential biases of any single segmentation backbone.

## A.6 Inference Time Analysis of VIDEOMOLMO vs. Molmo+SAM2

To assess the computational cost of our temporal modeling approach, we compare the inference time of VIDEOMOLMO with a baseline Molmo+SAM2 combination using a sampling rate of 4 and normalize the results per frame. VIDEOMOLMO incurs only a minimal additional overhead of 0.06 seconds per frame (0.81s vs. 0.75s), while achieving substantial performance gains of 9.0 percentage points on the MeViS (Ding et al., 2023) segmentation benchmark and 24 percentage points on counting tasks (VPoS-Bench). See Table 16

Table 16: Inference time comparison per frame (in seconds).

| Model | Inference time (s) |
|---|---|
| Molmo + SAM2 | 0.75 |
| VIDEOMOLMO | 0.81 |

## A.7 MODEL SIZE COMPARISON

We further compare the number of parameters, total GPU hours, and training efficiency of VIDEO-MOLMO against VideoLISA (Bai et al., 2024) and VideoGLAMM (Munasinghe et al., 2024). While VIDEOMOLMO requires 1000 GPU hours, it is $1.3\times$ faster than VideoLISA (Bai et al., 2024) (1280 GPU hrs) but $2.1\times$ slower than VideoGLAMM (480 GPU hrs). Despite having almost double the parameters compared to VideoGLAMM (Munasinghe et al., 2024) and VideoLISA (Bai et al., 2024), VIDEOMOLMO requires nearly the same total training time as VideoGLAMM (Munasinghe et al., 2024) (Table 17.

Table 17: Model size and training efficiency comparison.

| Model | Params (B) | GPU hrs | HW | Train Time (hrs) |
|---|---|---|---|---|
| VideoLISA | 4.4B | 1280 | 64×A10 (24GB) | 20 |
| VideoGLAMM | 4.4B | 480 | 4×A100 (40GB) | 120 |
| VIDEOMOLMO | 8.0B | 1000 | 8×A100 (80GB) | 125 |

Table 18: Comparison of mask→point vs. direct point prediction on the counting benchmark.

| Model | MAE ↓ | EMA ↑ |
|---|---|---|
| VideoGLaMM | 2.05 | 12.9 |
| VideoLISA | 2.43 | 20.0 |
| VIDEOMOLMO | **0.72** | **73.3** |

## A.8 GENERATING POINTS FROM SEGMENTATION MASKS

We analyze the effect of choosing a **point→mask** formulation over a **mask→point** formulation. While it is true that existing Video-LMMs can produce segmentation masks, converting these masks into points introduces several key limitations that motivate our direct pointing strategy.

To directly evaluate this alternative, we converted segmentation masks generated by state-of-the-art Video-LMMs like VideoLISA and VideoGLaMM into points using a standard centroid-based technique, as suggested by the reviewer. As shown in Table 18, our method significantly outperforms these baselines on the counting benchmark.

The mask-to-point approach breaks down particularly in challenging scenarios. When objects are in close proximity, segmentation models often yield merged or overlapping masks, making it difficult to localize individual instances. Additionally, for elongated or irregularly shaped objects, centroids may lie in background areas rather than on semantically meaningful parts of the object.

In contrast, our direct pointing strategy explicitly predicts fine-grained and semantically grounded spatial locations, allowing for more accurate and interpretable outputs. These results suggest that direct pointing not only simplifies the reasoning burden for the LMM but also constitutes a fundamentally different and more effective paradigm for fine-grained spatial understanding in videos.

## A.9 EFFECT OF BIDIRECTION TEMPORAL MASK FUSION MODULE AND SAMPLING RATE ON EFFICIENCY

We conducted an analysis to evaluate how the *Bidirectional Temporal Mask Fusion* module and the sampling rate affect the efficiency of VIDEOMOLMO.

Table 19: Per-frame processing time of VIDEOMOLMO at sampling rate $k = 4$.

| Pointing (s) | Post-processing (s) | Total (s) |
|---|---|---|
| 0.74 | 0.07 | 0.81 |

Table 20: Per-frame processing time of VIDEOMOLMO for different sampling rates $k$.

| Sampling Rate ($k$) | Time (s) |
|---|---|
| 4 | 0.81 |
| 8 | 0.48 |
| 10 | 0.35 |
| 15 | 0.31 |

**Bidirection Temporal Mask Fusion module overhead.** Table 19 reports the time taken by VIDEOMOLMO to process a video at a sampling rate of $k = 4$, normalized per frame. The pointing module requires approximately $0.74$ seconds per frame, while the post-processing module—which applies bidirectional mask fusion—adds a consistent $0.07$ seconds. Thus, the total processing time per frame is $0.81$ seconds, confirming that post-processing introduces only a small and stable overhead.

**Effect of sampling rate.** We further investigate how varying the sampling rate $k$ impacts efficiency. As shown in Table 20, increasing $k$ (i.e., processing fewer frames densely) reduces the overall per-frame computation time substantially. For example, raising $k$ from 4 to 15 decreases processing time from 0.81s to 0.31s per frame.

The post-processing time remains unaffected by the sampling rate because SAM2 must still segment all video frames, regardless of the sampling rate used for pointing.

### A.10 PERFORMANCE ON LARGE NUMBER OF OBJECTS

To evaluate VIDEOMOLMO performance on complex scenes, we tested on videos containing more than 30 objects. Table 21 shows that VIDEOMOLMO significantly outperforms both proprietary and open-source models on this challenging subset.

VIDEOMOLMO achieves the best results with an MAE of $0.86$ and EMA of $58.6\%$, substantially outperforming GPT-5 ($0.91$ MAE, $48.0\%$ EMA) and other baselines. Open-source models struggle considerably more, with Molmo showing the poorest performance ($1.45$ MAE, $39.5\%$ EMA). These results demonstrate that VIDEOMOLMO maintains strong counting and localization accuracy even in dense object scenarios where existing models fail.

### A.11 ADDITIONAL QUALITATIVE RESULTS

**General qualitative results.** We also present some qualitative results in Figures 8, **??**, 9, 10, and 11 on our proposed VPoS-Bench, MeViS, YT-VOS, Ref-DAVIS, and ReasonVOS, respectively. We observe that in each case, VIDEOMOLMO generates fine-grained points and corresponding masks pertaining to the query objects. In fact, VIDEOMOLMO performs well even in the cases of multi-object queries (¿2 objects) such as in VPoS-Bench counting task of Fig. 8 (1ˢᵗ row) and Fig. **??** (3ʳᵈ row). Further, VIDEOMOLMO also excels at grounding small and fine-grained objects. Fig. 8 (3ʳᵈ row) shows VIDEOMOLMO accurately points and grounds the far-away car on the road, although the car is too small to point at in some frames. Similarly, VIDEOMOLMO is able to ground the helmet in Fig. 10 (2ⁿᵈ row) while avoiding to ground the entire biker.

**Failure cases.** While VIDEOMOLMO demonstrates strong fine-grained pointing capabilities, it is not without limitations. As illustrated in Fig. 12, certain failure cases highlight areas for improvement. In the first row, the model is expected to point to the black harness but instead grounds a part of the adjacent bag. This misalignment stems from the limitations of SAM2, which struggles to accurately convert the predicted point coordinate into a meaningful mask. In the second row, the model points to only one of several visible paraglider lines, missing multiple lines. Such cases

Table 21: Performance of VIDEOMOLMO on videos with a large number of objects. (↓ lower is better, ↑ higher is better)

| Model | MAE ↓ | EMA ↑ |
|---|---|---|
| GPT-5 | 0.91 | 48.0 |
| Gemma3-12B | 1.15 | 34.6 |
| Qwen2.5-VL-7B | 1.02 | 40.0 |
| Molmo | 1.45 | 39.5 |
| VIDEOMOLMO | **0.86** | **58.6** |

suggest a need for enhanced expressiveness, such as enabling the model to predict multiple points for a single query. Addressing these limitations opens new avenues for future work in improving the robustness and granularity of point grounding in complex scenes.

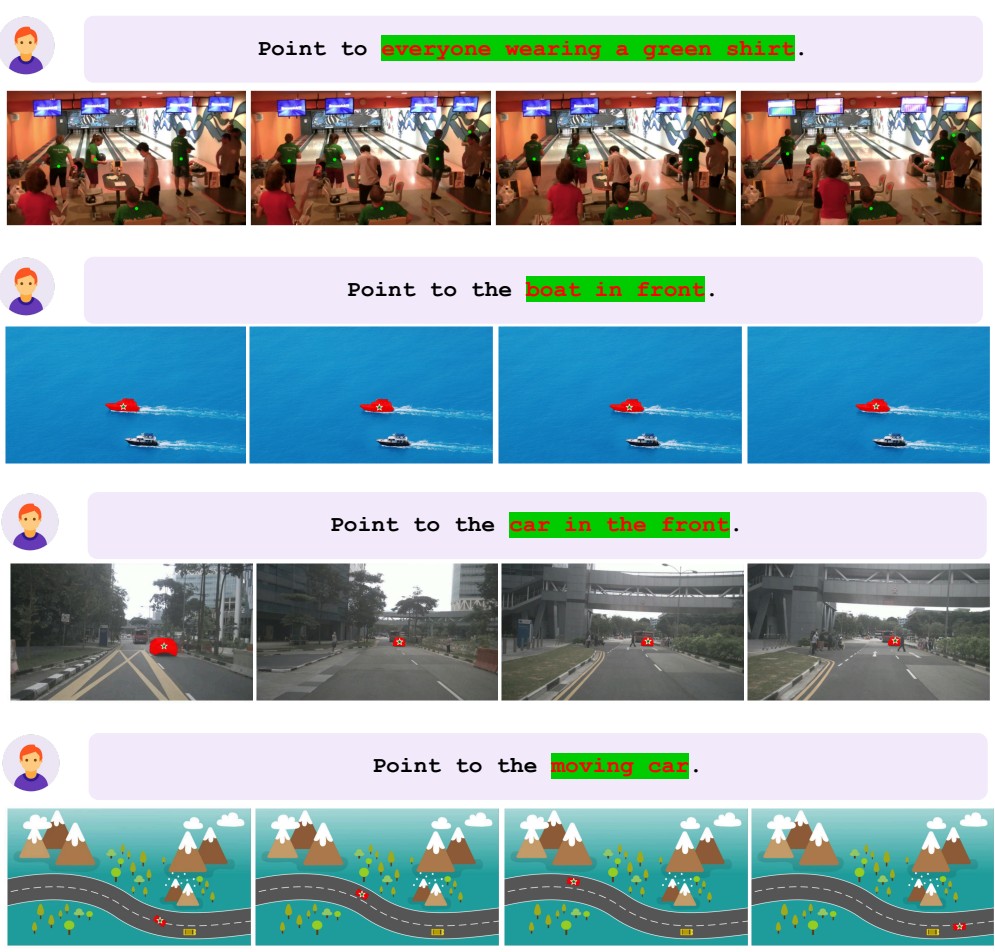

Figure 8: VPoS-Bench qualitative examples.

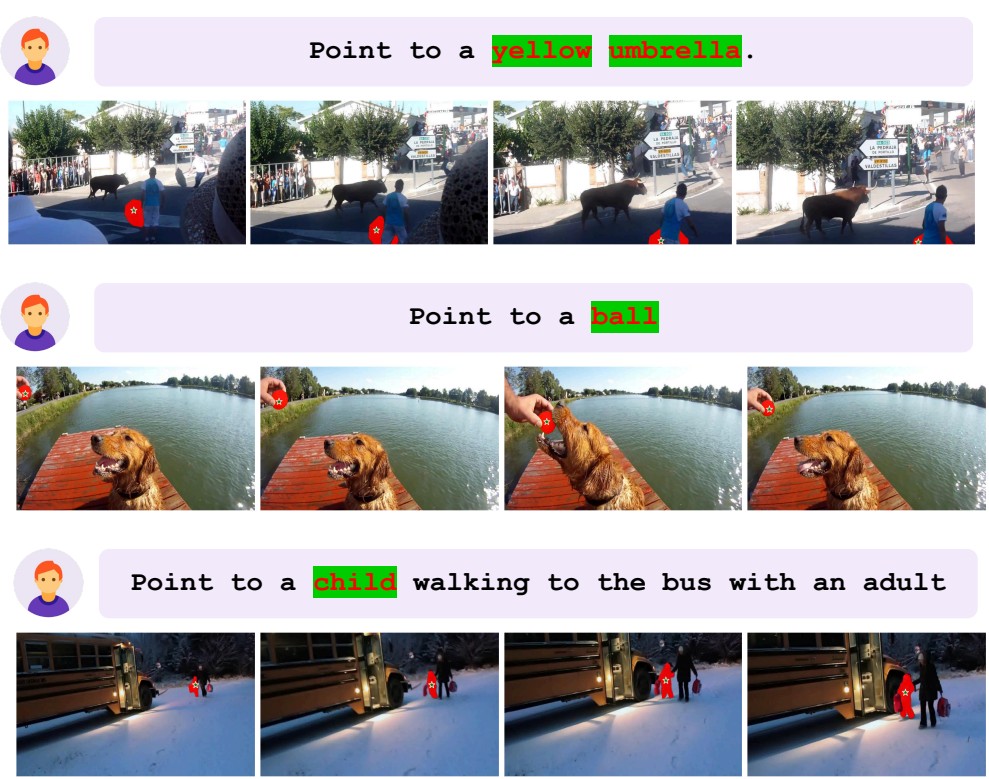

Figure 9: Refer-YouTube-VOS qualitative examples.

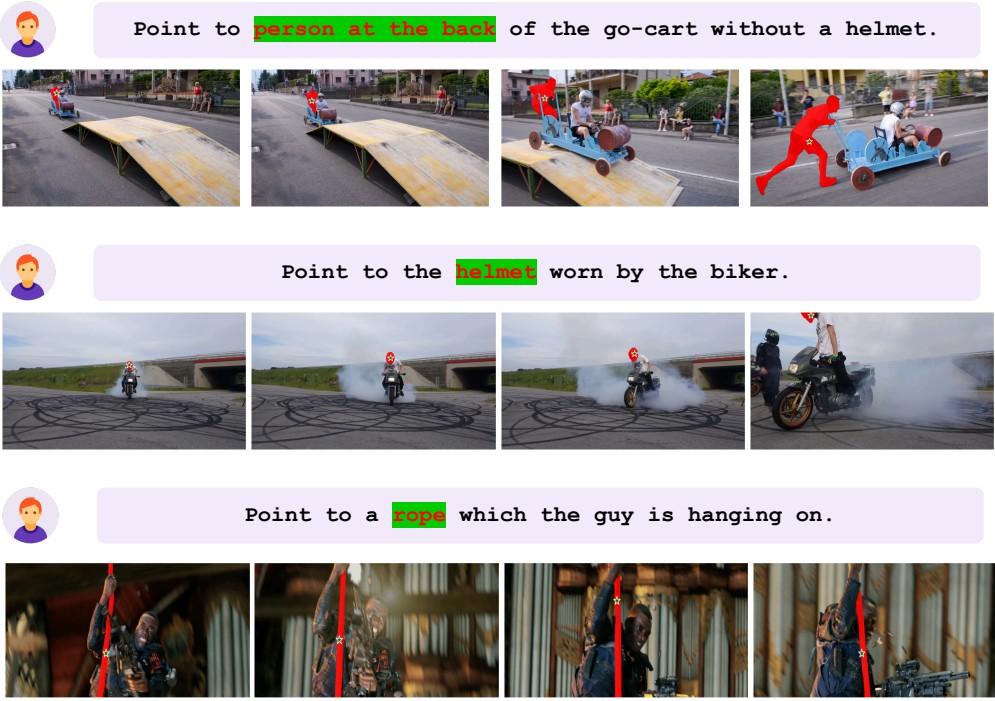

Figure 10: Refer-DAVIS qualitative examples.

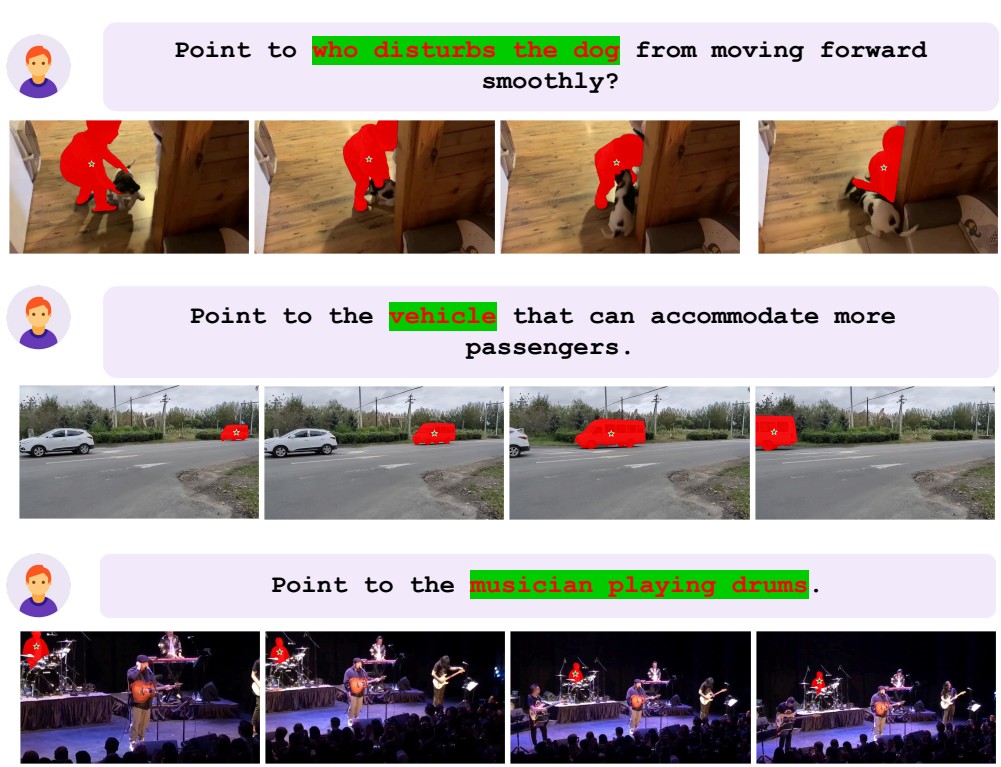

Figure 11: ReasonVOS qualitative examples.

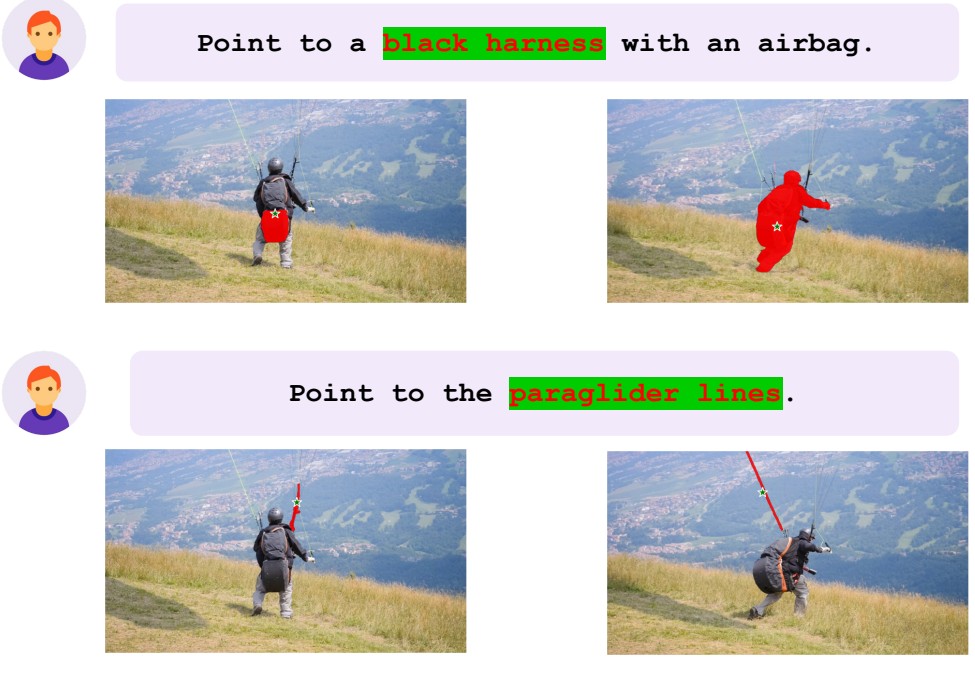

Figure 12: Qualitative failure cases of VIDEOMOLMO.