# OpenReview forum: "VideoMolmo: Spatio-Temporal Grounding meets Pointing"
_ICLR.cc/2026/Conference — ICLR 2026 Conference Withdrawn Submission_

### Official Review · Reviewer_unq8 · 2025-11-01

**Soundness:** 3
**Presentation:** 3
**Contribution:** 1
**Rating:** 2
**Confidence:** 5

**Summary:**

The paper addresses the challenge of fine-grained spatio-temporal localization by replacing dense mask prediction with point-based grounding, enabling interpretable and efficient localization across frames. Built on Molmo, it introduces 1) a temporal attention module for cross-frame reasoning and 2) a bidirectional temporal mask fusion mechanism for coherent point propagation for video understanding tasks.

**Strengths:**

- The proposed approach outperforms previous approaches on multiple datasets at multiple metrics.
- The evaluation benchmark is exhaustive on multiple aspects of spatio-temporal evaluation.

**Weaknesses:**

- Architecture Novelty
   - Section 4.1: Temporal Module: Aggregating information using past frames feature aggregation is a very common aspect of trivial video understanding. For video segmentation or dense tasks the understanding from frames to patch level aggregation. Earlier works utilize temporal feature aggregation or memory module - the idea is a base setup not a novelty. If there’s something missing I would like authors to clarify. Specifically, if there’s some previous work used as a baseline and then made changes for this paper.
   - Section 4.2: BiDirectional Temporal Mask Fusion: In the bidirectional propagation, are the masks interpolated for intermediate frames (between i and i+n)? The fusion strategy looks more like a heuristic based on masks - a hyperparameter value dependent. If the overlap is significant enough then take intersection otherwise union. This heuristic might be generalizable enough across datasets, but that doesn’t make a novel contribution. It can’t be a contribution to the ICLR level main conference paper.
- Dataset
   - VPoS-Bench: Since dataset generation is a contribution of the paper, I was expecting more stats or details about the dataset. The datasets combined all have different properties in terms of video length and query aspect. Can authors please go more in detail about the dataset? The range contains from simple scenarios to complex datasets such as MeVis. I tried to look at appendix, however, I couldn’t find the details.
- Result
    - The paper should include more approaches for fair comparison -> Table 3 [1].

[1] Ding, H., Tang, S., He, S., Liu, C., Wu, Z., & Jiang, Y. G. (2025). Multimodal referring segmentation: A survey. arXiv preprint arXiv:2508.00265.

**Questions:**

Please see the weakness section.

---

> ### Author Response · Authors · 2025-11-29
> **Official Comment by Authors**
>
> ### 1. Temporal Module
> We clarify that our temporal module is not a standard frame-level memory bank. While previous baselines typically aggregate temporal information by averaging or attending to whole-frame encodings, our approach is designed for dense, fine-grained patch-level reasoning.
>
> *Difference from previous works like STM, STCN, XMem* [1, 2, 3]: Because we build upon the static Molmo architecture (which uses a multi-crop strategy), simply averaging previous frames would destroy the fine-grained spatial details required for segmentation. Instead, we extract features for every patch within every crop, flattening them into a dense sequence $f \in \mathbb{R}^{(N \cdot P) \times D}$ (lines 207-210). These features are then reshaped into non-overlapping $2 \times 2$ windows ($\mathbb{R}^{(N\cdot P/4)\times 4\times D}$, lines 160-161), each containing four patch vectors. We then apply cross-attention across these patch windows. This makes our "temporal aggregation" actually a dense patch-to-patch attention mechanism. The current frame's patch window features (Query) explicitly attend to relevant spatial patch window features in the history (Key/Value), amplifying subtle motion cues and preserving the high-frequency details that a simple global aggregation methods typically would lose.
>
> We will update the manuscript to include these implementation details for greater clarity.
>
> ---
>
> ### 2. Bidirectional Temporal Mask Fusion
> Thank you for acknowledging the cross‑dataset generalizability of Bidirectional Temporal Mask Fusion. We use this module to convert the predicted points to segmentation masks to provide compatibility and consistent evaluation with previous works. Our position is that the community benefits when a system’s core reasoning (point‑first grounding with temporal context) is paired with the simplest effective mechanism needed for metric compatibility.
>
> (a) “*Are the masks interpolated for intermediate frames (between i and (i+n)?*”
> Kindly note that we do not interpolate masks. As stated in Section 4.2, we (i) convert the predicted points to masks on two sampled “anchor” frames using SAM2, then (ii) propagate these masks bidirectionally to each intermediate frame using SAM2’s video propagation (denoted  $Prop$ → Prop →  in Eq. (4)). This yields two candidate masks per intermediate frame—one from the past and one from the future—before fusion.
>
>
> (b) “*The fusion strategy is a heuristic (IoU threshold; intersection vs union), hence not a novel contribution.*":
> We agree the fusion rule is intentionally simple. Its purpose is systemic compatibility, not to be the centerpiece contribution. We redesign the problem around point‑first grounding and then provide a minimal, robust consensus step so that point predictions can be evaluated under mask metrics used by existing benchmarks. We will update the manuscript by explicitly stating that in the updated manuscript.
>
>
> (c )Ablation vs. alternatives (Table 5). On Refer‑DAVIS, our bidirectional IoU‑gated fusion reaches J\&F=72.45, outperforming naive choices such as “prefer left” (70.34, +2.11 pp), “prefer right” (69.37, +3.08 pp), “smaller mask” (65.64, +6.81 pp), and “intersection only” (61.89, +10.56 pp). Even against the stronger “larger mask” rule (71.65), we still see a +0.80 pp gain.
>
> (d) Generalizability / plug‑and‑play. When we add the same fusion to the Molmo+SAM2 baseline, it improves performance across all three datasets (Fig. 7), supporting the claim that the module is broadly useful and not tuned to our model.
>
> We hope this clarification resolves the concerns about Section 4.2 and helps situate the mask‑fusion module correctly within the paper’s broader contributions.
>
>
> [1] Oh, S. W., Lee, J.-Y., Xu, N., & Kim, S. J. (2019). Video Object Segmentation using Space-Time Memory Networks. In ICCV, 9226–9235.
>
> [2] Cheng, H. K., Tai, Y.-W., & Tang, C.-K. (2021). Rethinking Space-Time Networks with Improved Memory Coverage for Efficient Video Object Segmentation. In NeurIPS 2021.
>
> [3] Cheng, H. K., & Schwing, A. G. (2022). XMem: Long-Term Video Object Segmentation with an Atkinson–Shiffrin Memory Model. arXiv:2207.07115.

---

> > ### Author Response · Authors · 2025-11-29
> > **Continued Comment by Authors**
> >
> > ### 3. Details of Dataset
> > We apologize for the omission of detailed dataset statistics in the Appendix. We agree that understanding the composition of the training data is crucial, given that our dataset aggregates 72k video-caption pairs across diverse tasks ranging from standard referring segmentation to complex reasoning and long-term tracking.
> >
> > We will update the Appendix with the full statistics. Below, we provide the detailed breakdown used to construct our spatio-temporal pointing dataset:
> >
> > | Dataset | Domain | Video-Caption Pairs |
> > | :--- | :--- | :---: |
> > | **Reason-VOS** | Complex Reasoning and Logic | 24,885 |
> > | **MeViS** | Motion Expressions and Long-term | 23,051 |
> > | **Refer-YouTube-VOS** | Standard Referring VOS | 12,913 |
> > | **ViCaS-LGVIS** | Generative / Spatial Instructions | 6,813 |
> > | **GroOT** | Zero-shot / Open-world | 2,647 |
> > | **LaSOT** | Long-term Single Object Tracking | 1,120 |
> > | **Refer-DAVIS** | High-Quality VOS | 571 |
> > | **Total** | | **~72,000** |
> >
> > This composition was carefully chosen to balance short-term visual matching (Refer-DAVIS, Refer-YouTube-VOS) with temporal reasoning and long-term consistency (MeViS, LaSOT, Reason-VOS), ensuring the model is robust across varying video lengths and query complexities.
> >
> > ---
> >
> > ### 4. More Results
> > We have expanded our evaluation to include more approaches [1] in the below table. The results confirm that VideoMolmo remains highly competitive against recent state-of-the-art methods.
> >
> > Notably, VideoMolmo achieves the highest performance on the MeViS benchmark (53.9), outperforming the 26B-parameter Sa2VA model. This highlights our model's strength in handling complex temporal dynamics. Furthermore, we achieve consistent performance across all three benchmarks without relying on explicit dense mask supervision, distinguishing our approach from fully supervised baselines like VRS-HQ and Sa2VA.
> >
> >
> > | Model | Refer-DAVIS J\&F | Refer-YouTube-VOS J\&F | MeViS J\&F |
> > |-------|-----------------|------------------------|-----------|
> > | VideoLISA | 68.8 | 63.7 | 44.4 |
> > | VideoGLaMM | 69.5 | 66.8 | 45.2 |
> > | Molmo + SAM2 | 68.8 | 63.6 | 46.9 |
> > | DsHmp | 64.9 | 67.1 | 46.4 |
> > | ViLLa | 74.3 | 67.5 | 49.4 |
> > | SAMWISE | 68.5 | 67.2 | 48.3 |
> > | VRS-HQ-13B | 74.4 | 71.0 | 50.9 |
> > | Sa2VA-26B | **77.0** | 70.1 | 46.2 |
> > | RGA3-7B | 72.8 | 68.5 | 50.1 |
> > | MPG-SAM 2 | 72.4 | **73.9** | 53.7 |
> > | VIDEOMOLMO | 72.5 | 67.3 | **53.9** |
> >
> > [1] Ding, H., Tang, S., He, S., Liu, C., Wu, Z., & Jiang, Y. G. (2025). Multimodal referring segmentation: A survey. arXiv preprint arXiv:2508.00265.

---

### Official Review · Reviewer_MS2t · 2025-11-01

**Soundness:** 2
**Presentation:** 3
**Contribution:** 2
**Rating:** 4
**Confidence:** 4

**Summary:**

This work introduces VideoMolmo as a two-stage framework that grounds objects through point-based localization and a large-scale spatio-temporal pointing dataset of 72k video–caption pairs with 100k annotated points, and VPoS-Bench, a challenging benchmark spanning five real-world domains. Extensive experiments verify the effectiveness of the proposed framework and the benchmark.

**Strengths:**

1.The paper is well-written and easy to follow for readers.

2.A lot of quantitative experiments have been conducted to verify that VideoMolmo outperforms most previous state-of-the-arts across various downstream tasks.

**Weaknesses:**

1.One concern for this work is the motivation of the point-based grounding formulation. As mentioned in the manuscript, the point-level supervision is constructed from mask-level data, and it would be unclear why the point-based formulation would be better compared to other data formats such masks for visual grounding in videos? I think mask-based annotations can also transfer to various other forms like points, bounding boxes, etc. So the rationality of the point-based formulation of this work needs further clarifications and justifications.

2.There seem to be some missing details for the experimental comparisons. When comparing on the proposed VPoS-Bench, are the baselines like some video mllms retrained on the proposed spatio-temporal pointing dataset or just evaluated in a zero-shot manner? When comparing on other downstream tasks, is the proposed VideoMolmo trained from stratch on these downstream data or inherited with the pretrained knowledge of the proposed point-based dataset?

**Questions:**

Please refer to the weaknesses.

**Details Of Ethics Concerns:**

N/A.

---

> ### Author Response · Authors · 2025-11-29
> **Official Comment by Authors**
>
> ### 1. Motivation for using Points
>
> Thank you for raising an important point that we believe is central to our work. While it is true that ground-truth masks can be converted to points, our experiments show that *learning to predict points directly* is superior to predicting masks first and then converting them into points particularly in video settings. To clarify this, we break down our motivation into four key aspects:
>
> **(a) Cognitive Alignment**
> Our formulation is inspired by human visual attention, which prioritizes localizing an object's semantic center before processing fine-grained boundary details [1, 2]. By explicitly modeling this “pointing” mechanism, we disentangle core object representation from pixel-level boundary noise, yielding more robust grounding.
>
> **(b) The “Mask Merging” Problem**
> As shown in Appendix A.8 (Table 18), the mask-to-point approach (predict a mask → convert to center point) breaks down in crowded or occluded scenes. Segmentation models often merge neighboring instances into a single mask, causing the derived point to drift into the background or between objects.
> In contrast, our *direct point prediction* explicitly localizes individual instances, even in close proximity.
>
> **(c) Performance Benefits**
> We empirically validate this by comparing our method to mask-prediction baselines (VideoGLAMM, VideoLISA). The direct point formulation consistently yields higher accuracy in complex scenarios, showing that points are not a “downgrade” from mask, rather they provide a more stable anchor for temporal tracking.
>
> **(d) Anchors for Multiple Downstream Tasks**
> Points also provide strong priors for a broad set of downstream tasks, including referring segmentation, video object segmentation, and counting.
>
> We also analyze the effect of choosing a point→mask formulation over a mask→point formulation.converting the masks predicted by existing Video-LMMs into points introduces several key limitations that motivate our direct pointing strategy. (Section A.8)
>
> | Model        | MAE ↓ | EMA ↑ |
> |--------------|-------|--------|
> | VideoGLaMM   | 2.05  | 12.9   |
> | VideoLISA    | 2.43  | 20.0   |
> | **VIDEOMOLMO** | **0.72** | **73.3** |
>
>
> ---
>
> ### 2. Further Experimental Details
>
> **1. Baseline Evaluation on VPoS-Bench**
> All baselines are evaluated in a **zero-shot** manner using the official checkpoints released by the respective authors. We do *not* retrain or fine-tune these Video-LLMs on our dataset. Since mask-based baselines are inherently trained to produce segmentation masks, we evaluate them directly against our method.
>
> **2. VideoMolmo on Downstream Tasks**
> VideoMolmo is evaluated in a strictly **zero-shot** setting across all downstream tasks (Refer-DAVIS, Refer-YouTube-VOS, MeViS, VPoS-Bench). It is not fine-tuned on any of these datasets. Instead, it relies entirely on the generalizable representations learned from our large-scale pointing dataset.
>
> This demonstrates the effectiveness of our approach: VideoMolmo acts as a **generic, foundational model** capable of pointing, counting, and segmentation (via our bidirectional temporal propagation module) without requiring task-specific retraining. We will explicitly detail these settings in the revised experimental section.
>
> [1] Navon, D. (1977). Forest before trees: The precedence of global features in visual perception. Cognitive Psychology, 9, 353–383
>
> [2] Hegdé, J. (2008). Time course of visual perception: Coarse-to-fine processing and beyond. Progress in Neurobiology, 84(4), 405–439

---

### Official Review · Reviewer_uK38 · 2025-11-05

**Soundness:** 3
**Presentation:** 3
**Contribution:** 2
**Rating:** 4
**Confidence:** 3

**Summary:**

The paper introduces VideoMolmo, a two-stage video–language model:
It separates the task of identifying an object (by points) from other downstream tasks like segmenting the object.
The first stage predicts points that represent object identity.
The second stage uses these points to guide downstream models.

**Strengths:**

- Leveraging of pretrained segmentation models to generate instance masks based on point prompts

- Strong performance against baselines

**Weaknesses:**

- The assumption that points make it clear what the model means is wrong. The Segment Anything Model already shows that if one points at an eye, the pointing is ambiguous: it could be the eye, the head, or the whole body that is pointed at.

- Limited novelty: The paper uses known object-centric principles, such as disentangling object appearance from position. In this case, position is disentangled from the downstream task of mask generation. Additionally, the temporal averaging is a simple mean computation. Overall the model is a straightforward extension of the Molmo model to the video domain with on clever object centric trick.

- Use of synthetic point labels generated with SAM-V2 that are tuned specifically to generate high IoU when used with SAM-V2 but are not evaluated otherwise.

**Questions:**

- Why is the temporal aggregation just a simple average? Doesn’t that smooth away important temporal details?

- Are there any human-based evaluations showing that the generated annotations of the training sets are really meaningful?

---

> ### Author Response · Authors · 2025-11-29
> **Official Comment by Authors**
>
> ## 1. Points as a robust representation
>
> *Assumption that the point makes it clear what the model meant*:
> We agree that an isolated click is ambiguous; a point on an eye could refer to the eye, the head, or the whole bird. However, our approach fundamentally differs from standard interactive segmentation because our points are **language-conditioned**.
>
> For example, given a fine-grained query like **“eye of a pigeon”**, the predicted point is precisely localized to the specific part. If the query is **“pigeon”**, our semi-automatic data-generation pipeline ensures that the ground-truth point is placed centrally (e.g., on the torso) to maximize the IoU of the resulting mask. This resolves typical *part-vs-whole* ambiguity.
>
> **Handling Complexity:**
> Although a single point is sufficient in most cases, we discuss limitations in Appendix A.11 (Fig. 12). For objects where a single point is insufficient (e.g., elongated shapes), our model can predict **multiple points** to capture the full extent (Table 13).
>
> **Temporal Stability:**
> At inference, we use **bidirectional propagation** and **IoU-gated fusion** (Eqs. 4–6) to mitigate frame-wise inconsistencies, ensuring the point remains stable over time.
>
> We will revise the introduction to explicitly state that our points are interpretable because they are **“language-conditioned and supervised to maximize mask quality,”** distinguishing them from ambiguous raw clicks.
>
> ---
>
> ## 2. Temporal Module
>
> We clarify a key misunderstanding regarding the Temporal Module.
>
> ### 1. The temporal averaging is *not* a “simple mean computation.”
>
> As described in lines 207–210, our module is a **dense, windowed cross-attention mechanism**. Because we build on Molmo’s multi-crop architecture, a simple average would destroy the high-frequency spatial details necessary for segmentation.
>
> We extract features for every patch from every crop, flattening them into: $ f \in \mathbb{R}^{(N \cdot P) \times D} $. We then reshape them into non-overlapping $2 \times 2$ windows: $ f \in \mathbb{R}^{(N \cdot P / 4) \times 4 \times D} $ and apply **cross-attention** within each window.  The current frame (Query) attends to the relevant spatial windows in the history (Key/Value), enabling dense patch-to-patch temporal reasoning and amplifying subtle motion cues that global aggregation methods (e.g., memory banks in STM/XMem [1,2]) would lose.
>
> ### 2. Novelty of the “Object-Centric Trick”
>
> VideoMolmo is the first work to apply “disentangling position from appearance” via **point-centric reasoning** for Video-LMMs. This creates a **semantic bottleneck**, preventing mask-drift and merged-mask failures (Fig. 1). This disentanglement improves robustness in crowded scenes.
>
> ---
>
> ## 3. Generalizability of points learned by VideoMolmo
>
> We evaluated generalizability by pairing VideoMolmo with two architecturally different segmentation models: **FastSAM** [3] and **MobileSAM** [4] (Appendix A.5, Table 15).
>
> Although SAM-2 yields the strongest performance, these lightweight alternatives still retain **82–86%** of SAM-2’s performance.
>
> This demonstrates that VideoMolmo learns a **model-agnostic spatial coordinate**, and does not overfit to SAM-2. The model learns *where the object is*, independent of the segmentation backend.
>
> ---
>
> ## 4. Human Evaluation
>
> We randomly sampled 50-point annotations from the spatio-temporal VPoS dataset.
> Ten human judges were shown the video frame, the referring expression, and the synthetic point. Since each generated point lies within the object, judges could easily identify the intended target.
>
> They rated how well the point represents the intended object:
>
> | Rating | Interpretation | % of samples |
> |--------|----------------|--------------|
> | 1 | Weak | 3% |
> | 2 | Moderate | 11% |
> | 3 | High | 86% |
>
> These results show that most synthetic point annotations are judged as **highly representative**, indicating that the points are semantically meaningful rather than tailored to SAM-2 behavior. Importantly, judges were **not shown ground-truth masks**, ensuring that the evaluation reflects human perceptual understanding.
>
> [1] Oh, S. W., Lee, J.-Y., Xu, N., & Kim, S. J. (2019). Video Object Segmentation using Space-Time Memory Networks. In ICCV, 9226–9235.
>
> [2] Cheng, H. K., & Schwing, A. G. (2022). XMem: Long-Term Video Object Segmentation with an Atkinson–Shiffrin Memory Model. arXiv:2207.07115.
>
> [3] Zhao, X., Ding, W., An, Y., Du, Y., Yu, T., Li, M., Tang, M., & Wang, J. (2023). Fast Segment Anything. arXiv:2306.12156.
>
> [4] Zhang, C., Han, D., Qiao, Y., Kim, J. U., Bae, S.-H., Lee, S., & Hong, C. S. (2023). Faster Segment Anything: Towards Lightweight SAM for Mobile Applications. arXiv:2306.14289.

---

### Note · Authors · 2026-01-06

I have read and agree with the venue's withdrawal policy on behalf of myself and my co-authors.